# Variational Autoencoders with Decremental Information Bottleneck for Disentanglement

## Abstract

One major challenge of disentanglement learning with variational autoencoders is the trade-off between disentanglement and reconstruction fidelity. Previous incremental methods with only on latent space cannot optimize these two targets simultaneously, so they expand the Information Bottleneck while training to optimize from disentanglement to reconstruction. However, a large bottleneck will lose the constraint of disentanglement, causing the information diffusion problem. To tackle this issue, we present a novel decremental variational autoencoder with disentanglement-invariant transformations to optimize multiple objectives in different layers , termed DeVAE, for balancing disentanglement and reconstruction fidelity by decreasing the information bottleneck of diverse latent spaces gradually. Benefiting from the multiple latent spaces, DeVAE allows simultaneous optimization of multiple objectives to optimize reconstruction while keeping the constraint of disentanglement, avoiding information diffusion. DeVAE is also compatible with large models with high-dimension latent space. Experimental results on dSprites and Shapes3D that DeVAE achieves a good balance between disentanglement and reconstruction. DeVAE shows high tolerant of hyperparameters and on high-dimensional latent spaces.

## 1 Introduction

Unsupervised learning for sensing the properties of objects is crucial to reduce the gap between humans and machines intelligence. Inline with human intelligence disentanglement learning (Bengio et al., 2013) is considered to be a promising direction to obtain explanatory representations from observations to understand and reason objects without any supervision.

In the recent years, various approaches (Higgins et al., 2017; Chen et al., 2018; Kim & Mnih, 2018; Burgess et al., 2018; Chen et al., 2016) have been proposed to successfully extract basic properties of objects, such as position, color, orientation, and scale (Burgess & Kim, 2018; Matthey et al., 2017). The commonly-used methods are based on variational autoencoder (VAE) (Kingma & Welling, 2014). In particular, $\beta$-VAE (Higgins et al., 2017) introduced an extra parameter $\beta$ on the Kullback-Leibler (KL) divergence to promote disentanglement. However, there is a trade-off between disentanglement and reconstruction fidelity on $\beta$-VAE, which is a problem to be solved in the following works.

One common direction for dealing with the trade-off is to penalize the Total Correlation (TC) between latent variables, avoiding reducing the mutual information, such as FactorVAE (Kim & Mnih, 2018), $\beta$-TCVAE (Chen et al., 2018), and DIPVAE (Kumar et al., 2018). As pointed out in (Träuble et al., 2020; Dittadi et al., 2020), TC-based VAEs have a strong prior assumption that the factors are statistically independent. Beyond that, when it comes to high-dimension latent space, the estimation of TC becomes inaccurate due to the curse of dimensionality, as our experiments observed in Section 3.2. The realistic problems usually have numerous factors, therefore it would need a large model with high latent space to extract representations. For example, the popular deep model ResNet50 (He et al., 2016) has 2048 dimensional feature space. However, the current TC estimations are not scaled to high dimensional problems, causing the low performance of BC-based methods in practice. In this work, instead of calculating TC, we leverage the information bottleneck (IB) (Tishby et al., 1999; Burgess et al., 2018) to promotes disentanglement.

In the meanwhile, previous information bottleneck (IB)-based methods (Burgess et al., 2018; Shao et al., 2022; Wu et al., 2022) have tried to solve the obstacle of trade-off between disentanglement and reconstruction fidelity. A narrow IB enforces the model to find efficient codes for representing the data, which encourages disentanglement. Therefore, they first set a high pressure with a narrow IB and then expand the IB gradually to promote disentanglement to reconstruction fidelity , termed *incremental methods*. For example, DynamicVAE (Shao et al., 2022) initiated $\beta$ with a large value at the beginning of training for disentanglement and stably increase the KL divergence for reconstruction by a non-linear PI controller (Åström & Hägglund, 2006). However, they lost the constraint of disentanglement when expanding the IB, which causes the information diffusion problem (Wu et al., 2022). In this work, to avoid information diffusion, we aim to optimize reconstruction while keeping the constraint of disentanglement.

Different from IB-Incremental based approaches listed above, our key motivation is to optimize disentanglement and reconstruction simultaneously. revious methods only have one latent space and are unable to optimize disentanglement and reconstruction at the same time, which causes them to have to change the target from disentanglement to reconstruction during training. Instead, our work proposes a novel multi-layer framework with its own latent spaces and objectives in each layer, allowing optimizing multiple targets at a time. In this way, the first layer is a vanilla VAE to rebuild high-quality images, and the subsequent layers will distill some important variables by narrow IBs to promote disentanglement. To inherit disentanglement from the subsequent layers, we introduce **disentanglement-invariant transformations** to connect the layers one by one. These extra layers can be seen as regularizations for disentanglement to constrain the representation.

To achieve this, we propose a simple yet effective VAE framework composed of multiple continuous latent sub-spaces with a novel IB-Decremental strategy and disentanglement-invariant transform operators, which we call DeVAE. Specifically, we decrease the information bottleneck of each latent space layer by layer, where we constrain the first space for informativeness to recover the input image, and other disentangled spaces for learning factors of the image by narrow IBs. Furthermore, we introduce the disentanglement-invariant transform operator to ensure simultaneous optimization of disentanglement across continuous latent sub-spaces, which avoids the information diffusion. Our decremental model avoids ID by keeping the constraints of disentanglement while optimizing reconstruction. We also conducted comprehensive comparisons with popular methods quantitatively and qualitatively. Experimental results demonstrate that DeVAE is robust in hyperparameters and the size of latent spaces.

Our contributions can be summarized as follows:

- We introduce several latent spaces sharing disentanglement by disentanglement-invariant transformations.

- We propose a novel diagram for disentanglement learning by decreasing IB, termed decremental VAE (DeVAE). Our decremental model can handle large-scale problems and show robustness on several datasets.

## 2 METHODOLOGY

### 2.1 PRELIMINARIES

**Problem Setup & Notations.** Disentanglement learning aims to learn the factors of variation which raises the change of observations. Given a set of samples $x \in \mathcal{X}$, they can be uniquely described by a set of ground-truth factors $c \in \mathcal{C}$. Generally, the generation process $g(\cdot)$ is invisible $x = g(c)$. We say a representation for factor $c_i$ is disentangled if it is invariant for the samples with $c_j$. We use variational inference to learn the disentangled representation for a given problem. $p(z|x)$ denotes the probability of $z = f(x)$, $p(x|z)$ denotes the probability of $x = g(z)$. The representation function is a conditional Bayesian network of the form $q_\phi(z|x)$ to estimate $p(z|x)$. The generative model is another network of the form $p_\theta(x|z)p(z)$. $\phi, \theta$ are trainable parameters.

**Revisit VAE & $\beta$-VAE.** The VAE framework (Kingma & Welling, 2014) computes the representation function by introducing $q_\phi(z|x)$ and optimizing the variational lower bound (ELBO).

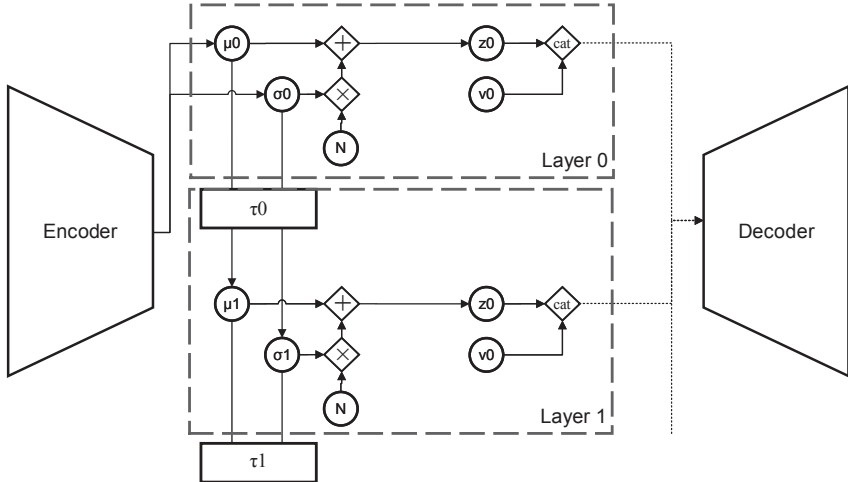

Figure 1: Illustration of our Decremental Variational Autoencoder (DeVAE). The solid lines denote the information flow of the encoding process. The dash lines denote the decoding process which randomly selects one layer's representation and concatenates the corresponding embedding vector. $v_i$ denotes a layer embedding. $\tau_i$ denotes a disentanglement-invariant transformation. $\boldsymbol{\mu}_i, \boldsymbol{\sigma}_i$ denote the parameters of latent variables $\boldsymbol{z}_i$. $N$ denotes a random noise. Each layer has a pressure $\beta_i$ to control the capacity of IB.

$\beta$-VAE (Higgins et al., 2017) introduces the hyperparameter $\beta$ to control the IB:

$$\mathcal{L}(\theta, \phi) = \mathbb{E}_{q_\phi(\boldsymbol{z}|\boldsymbol{x})}[\log p_\theta(\boldsymbol{x}|\boldsymbol{z})] - \beta D_{\mathrm{KL}}(q_\phi(\boldsymbol{x}|\boldsymbol{z})||p(\boldsymbol{z})). \tag{1}$$

Consider using $\beta$-VAE to learn a representation of the data; the representation will be disentangled but lose information when $\beta$ is large (Burgess et al., 2018). We can set a large $\beta$ to learn a disentangled representation and a small $\beta$ to learn an informative representation.

However, previous disentanglement methods (Higgins et al., 2017; Chen et al., 2018; Burgess et al., 2018) are limited in low-dimension latent space and poorly deal with the trade-off between disentanglement and reconstruction. Current state-of-the-art approach (Shao et al., 2022) with an annealing manner from high pressure to low pressure will loosen the constraint of disentanglement when reducing the pressure. To address this issue, we propose a novel decremental variational autoencoder with hierarchical latent spaces, namely DeVAE, to optimize disentanglement and reconstruction fidelity simultaneously, which can handle high-dimensional latent spaces, as shown in Figure 1. Our DeVAE applies a hierarchical structure with a decremental information bottleneck and disentanglement-invariant transformation to produce latent variables layer by layer. The decoder part randomly selects one layer's latents concatenating an embedding vector to generate images.

## 2.2 Hierarchical Latent Spaces with Decremental Information Bottleneck

In order to retain the disentanglement constraint while optimizing the reconstruction fidelity, we introduce a Hierarchical Latent Space (HiS) with $K$ layers and assign a pressure $\beta_i$ for the $i$th layer $\mathcal{Z}_i$ to promote disentanglement.

The first layer aims to rebuild the dataset and uses the ELBO as objective. The subsequent layers will promote disentanglement by reducing the IB. Therefore, the objective of the $i$th layer is

$$\mathcal{L}_i(\theta, \phi) = \mathbb{E}_{q_\phi(\boldsymbol{z}_i|\boldsymbol{x})}[\log p_\theta(\boldsymbol{x}|\boldsymbol{z}_i, \boldsymbol{v}_i)] - \beta_i D_{\mathrm{KL}}(q_\phi(\boldsymbol{z}_i|\boldsymbol{x})||p(\boldsymbol{z}_i)), \tag{2}$$

where $\boldsymbol{v}_i \in \mathbb{R}^{1 \times D}$ denotes the learnable layer embedding for the $i$th layer, $p_\theta(\boldsymbol{x}|\boldsymbol{z}_i, \boldsymbol{v}_i)$ is the decoder network shared with all layers, $q_\phi(\boldsymbol{z}_i|\boldsymbol{x})$ is the encoder network dependent on previous ones, $\beta \in \mathbb{R}^K$ is a set of coefficients to penalize the IB, particularly $\beta_0 = 1$. The parameters of each layer

are parameterized as a bottom-up process:

$$q(\boldsymbol{z}_i|\boldsymbol{x}) = \mathcal{N}(\mu_i(\boldsymbol{x}), \sigma_i(\boldsymbol{x})^2),$$
$$q(\boldsymbol{z}_0|\boldsymbol{x}) = \mathcal{N}(\mu_0(\boldsymbol{x}), \sigma_0(\boldsymbol{x})^2), \tag{3}$$
$$\mu_i(x), \sigma_i(x) = \tau_i(\mu_{i-1}(\boldsymbol{x}), \sigma_{i-1}(\boldsymbol{x})), i > 0$$

where the first layer $q_\phi(\boldsymbol{z}_0|\boldsymbol{x})$ is a conditional Bayesian network, $\tau_i$ denotes a transformation to modify the poster distribution of the previous layer to fit the layer objective $\mathcal{L}_i(\theta, \phi)$.

According to information theory, information can only decrease while processing, therefore *we gradually decrease the IB in the sequential layers, i.e., $\beta_{i+1} > \beta_i$.* In this way, the last layer with a narrow IB can promote disentanglement only, and the reconstruction fidelity will become better and better from the bottom to the top.

## 2.3 DISENTANGLEMENT-INVARIANT TRANSFORMATION

Though we create multiple latent spaces, these objectives only encourage the local representations to be disentangled or informative. We need a mechanism to connect these objectives for balancing disentanglement and reconstruction in one layer. In order to make sure disentanglement across all latent layers, we propose a disentanglement-invariant transformation (DiT) denoted as $\tau$.

**Theorem 1** $w \cdot \boldsymbol{z}$ *is disentangled if $\boldsymbol{z}$ is disentangled, $w$ is a diagonal matrix.*

Proof in Appendix A.2.

According to Theorem 1, we can scale the latent space to keep disentanglement. Scaling the posterior $\boldsymbol{z}_i$ violates the generation process which wants the marginal distribution $q(\boldsymbol{z}) = \sum q_\phi(\boldsymbol{z}|\boldsymbol{x})p(\boldsymbol{x})$ to be close to a standard normal distribution. Besides, most downstream tasks use the mean representation instead of sampled representation. Therefore, we only need the mean representations disentanglement-invariant.

The disentanglement-invariant transformation scales the parameters of the $i$th layer:

$$\tau_{\boldsymbol{w}^1, \boldsymbol{w}^2}(\boldsymbol{\mu}, \boldsymbol{\sigma}) = h(\boldsymbol{w}^1)\boldsymbol{\mu}, h(\boldsymbol{w}^2)\boldsymbol{\sigma}, \tag{4}$$

where $\boldsymbol{w}^1, \boldsymbol{w}^2$ are learnable diagonal matrices belonging to the $i$th layer, $h(w) = e^w$ is the power function to make sure the scaling values greater than 0. Therefore, we get the parameters of $i$th latent variables

$$\boldsymbol{\mu}_i = h(\sum_{j=0}^{i-1} w_j^1)\boldsymbol{\mu}_0, \quad \boldsymbol{\sigma}_i = h(\sum_{j=0}^{i-1} w_j^2)\boldsymbol{\sigma}_0, \quad i > 0. \tag{5}$$

and the $i$th KL divergence

$$D_{\mathrm{KL}_i} = \frac{1}{2}(1 + 2\sum_{j=0}^{i-1} w_j^2 + 2\log(\boldsymbol{\sigma}_0) - h(2\sum_{j=0}^{i-1} w_j^2)\boldsymbol{\sigma}_0^2 - h(2\sum_{j=0}^{i-1} w_j^1)\boldsymbol{\mu}_0^2). \tag{6}$$

## 2.4 OPTIMIZATION ALGORITHM

In this section, we combine the above components and introduce the optimization algorithm for the multiple objectives. We use a random process to optimize one layer's objective from $K$ latent spaces:

$$\mathcal{L}(\theta, \phi) = \mathbb{E}_{p(\boldsymbol{z}_i)}[\mathcal{L}_i(\theta, \phi)], \tag{7}$$

where $p(\boldsymbol{z}_i)$ denotes the probability of optimizing the $i$th latent space $\boldsymbol{z}_i$, which is defined as:

$$p(\boldsymbol{z}_i) = \begin{cases} \frac{1}{K}, & \text{for } s = 1 \\ (1 - s)\dfrac{s^i}{1 - s^{i+1}}, & \text{for } s > 1 \end{cases} \tag{8}$$

where $s$ denotes the power annealing of scale hyper-parameter for each $p(\boldsymbol{z}_i)$. In experiments, we empirically find that $s = 1$ achieves better performance, as observed in Section 3.3. Note that we do

```
1  def loss_fn(x,encoder,decoder,W,embeddings,betas,K):
2      idx = np.random.randint(K)
3      mu, logvar = encoder(x)
4      for i in range(idx):
5          w1, w2 = W[i]
6          mu = torch.exp(w1[i]) * mu
7          logvar = logvar + w2[i]
8      z = sample(mu, logvar) # re-parameter trick
9      recon = decoder(torch.cat([z,embeddings(idx)],1))
10     loss = F.mse(recon,x) + betas[idx] * kld(mu,logvar)
11     return loss
```

Algorithm 1: PyTorch-like implementation of DeVAE loss.

not aggregate the objectives into a loss, instead, DeVAE only rebuilds the images and optimizes the objective of one layer in one mini-batch.

In our model, $q_\phi(z|x)$ and decoder $p_\theta(x|z)$ are modelled by two neural networks, a $K$-size sequence 'betas' denotes the penalties on the KL divergences of corresponding layers, $w^1, w^2$ stores the learnable parameters for transforming latent spaces. First, we randomly sample a mini-batch and choose a target layer to optimize. Then use the algorithm introduced in Section 2.3 to obtain the representation of the target layer and reconstruct the corresponding images. Instead of using $K$ separated decoders to rebuild images, we apply a shared decoder with layer embeddings to reduce the parameter size. The only extra computational cost comes from $w^1, w^2$ and layer embeddings. Therefore, the parameter size and overhead are similar to the vanilla VAE. The PyTorch-like algorithm is shown in Algorithm 1.

## 3 EXPERIMENTS

### 3.1 EXPERIMENTAL SETUP

**Datasets.** We evaluate our method on two widely-used datasets (dSprites, Shapes3D). dSprites (Matthey et al., 2017) has 737,280 binary 64 × 64 x 1 images generated from five factors: shape (5), orientation (40), scale (6), position X (32), and position Y (32). Shapes3D (Burgess & Kim, 2018) has 480,000 RGB 64 × 64 × 3 images of 3D shapes generated from six factors: floor color (10), wall color (10), object color (10), object size (8), object shape (4), and azimuth (15).

**Evaluation Metrics.** We apply the following metrics to evaluate the performance of disentanglement and reconstruction. **MIG** (Chen et al., 2018): the mutual information gap between two variables with the highest and the second-highest mutual information. **FactorVAE metric** (Kim & Mnih, 2018): the error rate of the classifier, which predicts the latent variable with the lowest variance. **DCI Dis.**: abbreviation for DCI Disentanglement (Eastwood & Williams, 2018), a matrix of relative importance by regression. **Recon.**: abbreviation for Reconstruction Error, a measure of the distance between images; we use Mean Squared Error for RGB images and Binary Cross Entropy for binary images.

**Implementation.** We use a convolutional neural network as the encoder and a deconvolutional neural network as the decoder. Detailed architecture can be found in Appendix A.1. The activation function is ReLU. The optimizer is Adam (Kingma & Ba, 2015) with a learning rate of 1e-4, $\beta_1 = 0.9$, $\beta_2 = 0.999$. The batch size is 256, which accelerates the training process. All experiments train $300,000$ iterations by default. For the hyperparameters, we set $\beta = 12$ for $\beta$-TCVAE, $\beta = 6$ for $\beta$-VAE, and $K_i = 0.001, K_p = 0.01$ for DynamicVAE. According to the information freezing points (IFP) (Wu et al., 2022), beta=40 can filter factors orientation and shape, beta=10 can only filter factor orientation, so we set $\{\beta_i\} = [1, 10, 40], s = 1$ for DeVAE.

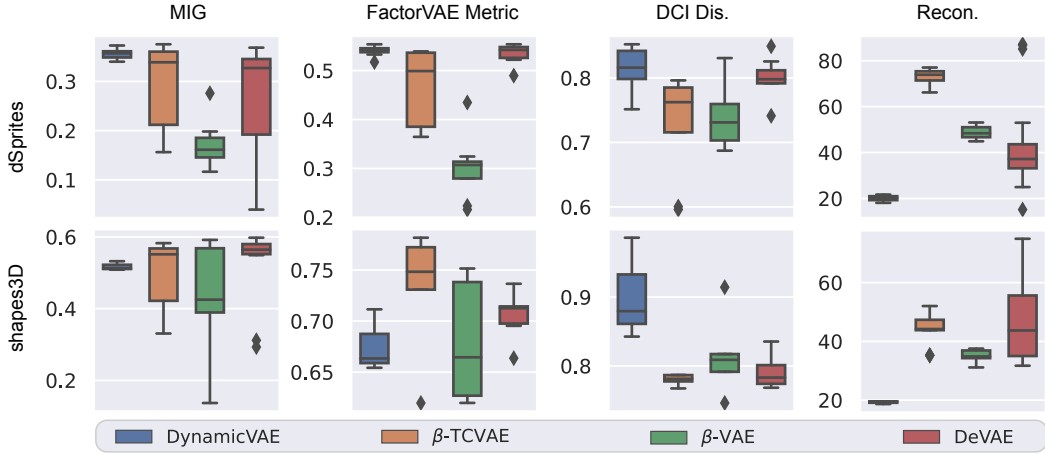

Figure 2: Box plots of quantitative benchmarks MIG, FactorVAE, Disentanglement, and reconstruction error on dSprites and Shapes3D.

## 3.2 COMPARISON TO PRIOR WORK

To demonstrate the effectiveness of the proposed DeVAE, we compare it to previous all types of baselines: 1) $\beta$-VAE (Higgins et al., 2017): the popular method for disentanglement and also the baseline model for DeVAE when there is only one latent layer; 2) $\beta$-TCVAE (Chen et al., 2018): the TC-based method with a good balance of simplicity and effectiveness; 3) Dynamic-VAE (Shao et al., 2022): the latest method with incremental information bottleneck.

**Disentanglement & Reconstruction.** We conducted experiments on dSprites and Shapes3D to compare the above methods. Each trail was repeated 10 times with different random seeds. We draw the distributions of three disentanglement scores and reconstruction errors in Figure 2. The visualization of sampling from the best models is shown in Appendix A.5. Experimental results reveal that DeVAE achieves an average improvement of 8% comparing to $\beta$-TCVAE and 47% to $\beta$-VAE on dSprites for disentanglement. DeVAE has a lower average reconstruction error on two datasets by 2% for $\beta$-TCVAE and by 30% for $\beta$-VAE. Though the improvement is not significant, $\beta$-TCVAE and $\beta$-VAE are not robust to one hyperparameter setting. Though DynamicVAE achieves the best overall results, it still suffers from ID problems and is incapable of dealing with high-dimensional space, see Figure 5 and Appendix A.7.

**Qualitative Visualization.** We also conducted a qualitative analysis to assess disentanglement. R3Q4We show the selected latent traversals whose KL divergence is larger than 0.5 in Figure 3. We can see that DeVAE disentangles position X and position Y perfectly. Shape, scale, and orientation are hard to be disentangled. We show the latent traversals of the best models with the highest MIG in Appendix A.4.

**Preventing Information Diffusion.** Information diffusion is a phenomenon of disentangling that one factor's information diffuses into other latent variables while training, causing the disentanglement scores to fluctuate during training (Wu et al., 2022). We hypothesize that losing the constraint of disentanglement is the reason for ID.

To prove it, we monitored the changes in mutual information during training. From Figure 4, we see that DynamicVAE has a significant trend of losing information on iteration 3e5. It means that the learned structure of representation was destroyed when expanding the IB. In contrast, DeVAE shows a relatively steady trend of increasing information for consistent regularizing. DeVAE overcomes the drawback of traditional information bottleneck-based methods by keeping the constraint of disentanglement.

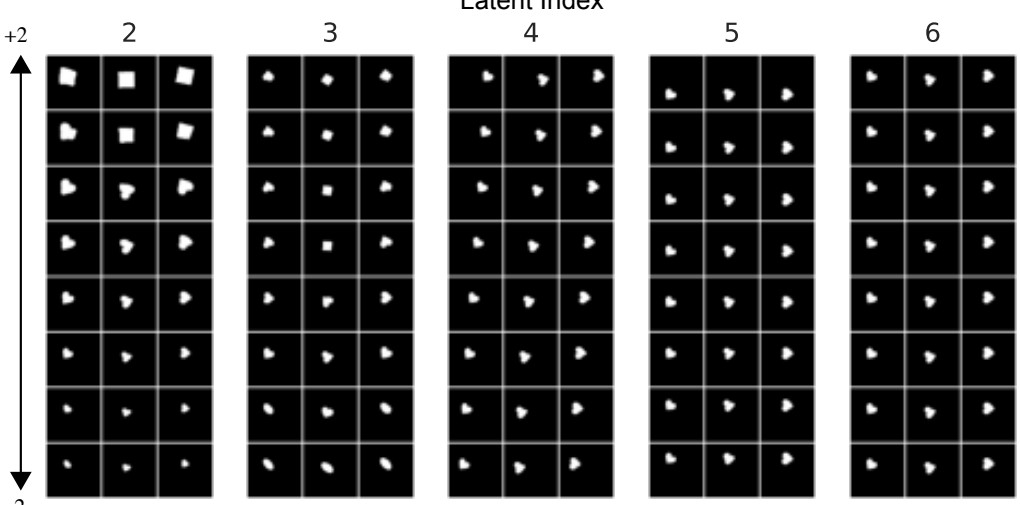

Figure 3: Latent traversal of DeVAE with the best MIG score on dSprites. Each block shows the generated images of traversing the latent variable from $-2$ to 2. Each group shows the traversal images sampling from 3 different random noise. The title above each group denotes the index of latent variable.

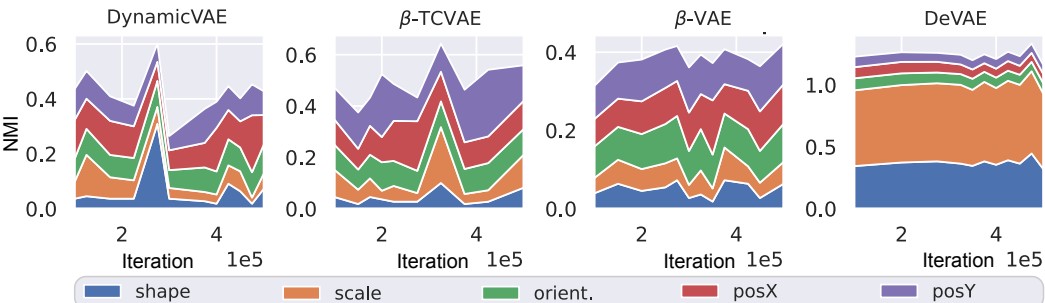

Figure 4: Comparison results of information diffusion. Each colored curve denotes the learned information that belongs to one factor over training iterations.

**Scaling to High-dimensional Latent Space.** Most disentanglement methods evaluate their performance on simple scenes with only one object and few factors. It is a challenge to extend these methods to complex scenes. However, whether these methods adapt to a large latent space to fit more factors is questionable. In particular, the dimension of latent space affects the estimation accuracy of MI for the TC-based methods.

To study the effect of high-dimensional latent space on estimating TC, we first generate samples from a D-dimensional multi-variable normal distribution $\boldsymbol{x}$ which is divided into two groups $\boldsymbol{x}^1$ and $\boldsymbol{x}^2$ with D/2 dimensions. The variables in a group are independent $\text{Cov}(\boldsymbol{x}_i^m, \boldsymbol{x}_j^m) = 0, i \neq j$; the variables between groups are correlative $\text{Cov}(\boldsymbol{x}_i^m, \boldsymbol{x}_i^n) = \rho, m \neq n$; each variable is a standard normal distribution. So, the TC of $\boldsymbol{x}$ is

$$\text{TC}(\boldsymbol{x}) = -\frac{\text{D}}{4}\log(1 - \rho^2). \tag{9}$$

We trained a discriminator for 2000 iterations to estimate the TC introduced in FactorVAE (Kim & Mnih, 2018). We compared the estimated TC and the real TC over dimensions and $\rho$. Each trail was repeated 10 times, and we report the average results as shown in Table 1. One can see that increasing $\rho$ or dimension diminishes the accuracy of estimation, and the estimators always have low errors when the dimension is 10. However, the estimation becomes extremely inaccurate when

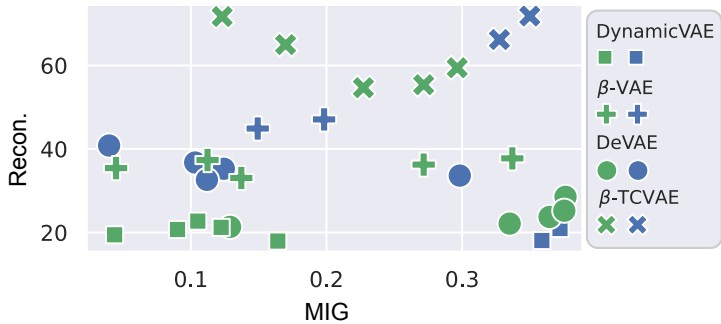

Figure 5: Distributions of MIG scores and reconstruction errors for low-dimensional space (blue) and high-dimensional space (green). The points in the bottom right have a better balance of disentanglement and reconstruction.

Table 1: The estimated MI of FactorVAE and the real MI on high dimensional spaces. The cases having large error are bold. $\rho$ denotes the correlation of two random variables.

| Dim | $\rho$ | Estimated TC | TC | Error |
|---|---|---|---|---|
| 10 | 0.3 | 0.23 | 0.24 | 0.03 |
| | 0.6 | 1.08 | 1.12 | 0.03 |
| | 0.9 | 4.14 | 4.15 | 0.00 |
| 100 | 0.3 | 2.17 | 2.36 | 0.08 |
| | 0.6 | 10.27 | 11.16 | 0.08 |
| | 0.9 | 23.39 | 41.52 | 0.44 |
| 1000 | 0.3 | 8.62 | 23.58 | 0.63 |
| | 0.6 | 17.40 | 111.57 | 0.84 |
| | **0.9** | **22.47** | **415.18** | **0.95** |

the dimension raises to 1000, which means such estimation will fail to penalize the TC for large models.

We further conduct experiments on dSprites to validate the above conclusion. The experimental settings are the same except for increasing the dimension of latent space to 1024. From Figure 5, we can see that $\beta$-TCVAE and DynamicVAE have significant performance decay. Higher dimensional space increases the complexity of calculating the TC and leads to significant estimation errors and also increases the chance of the ID problem for DynamicVAE see in Appendix A.7. $\beta$-VAE and DeVAE show robustness in high-dimensional latent spaces, which is necessary to train a large model on large data.

### 3.3 EXPERIMENTAL ANALYSIS

In this section, we performed ablation studies on the benefit of the proposed Hierarchical Latent Spaces (HiS) and Disentanglement-invariant Transformation (DiT). We also conducted extensive experiments to explore the effect of $\beta$ and $s$ on disentanglement and reconstruction performance.

**Hierarchical Latent Spaces & Disentanglement-invariant Transformation.** To demonstrate the effectiveness of the introduced Hierarchy Latent Spaces (HiS) and Disentanglement-invariant Transformation (DiT), we performed ablation experiments on the following situations: 1) The model has one single latent space; 2) The model applies a parallel structure instead of the hierarchy that latent spaces are independent; 3) We replace DiT with Linear Transformation ($\tau_i(z_i) = wz_i$), $w$ is an arbitrary matrix; 4) The proposed model DeVAE.

We report the MIG and Recon. for each layer in Table 2. MS and HiS can optimize multiple objectives for these layers separately. DiT enforces all layers to share disentanglement. In this

Table 2: Ablation Study on Multiple Spaces (MS), Hierarchical Structure (HiS) and Disentanglement-invariant Transformation (DiT). The reconstruction fidelity becomes better and better from the bottom layer to the top layer, but the disentanglement gets worse. DiT can prevent disentanglement to degenerate.

| MS | HiS | DiT | MIG | | | Recon. | | |
|----|-----|-----|--------|--------|--------|--------|--------|--------|
| | | | Layer0 | Layer1 | Layer2 | Layer0 | Layer1 | Layer2 |
| ✗ | ✗ | ✗ | 0.19 | - | - | 47.49 | - | - |
| ✓ | ✗ | ✗ | 0.24 | 0.32 | **0.35** | **22.21** | 40.79 | 62.40 |
| ✓ | ✓ | ✗ | 0.24 | 0.29 | 0.30 | 38.82 | 45.48 | 63.78 |
| ✓ | ✓ | ✓ | **0.35** | **0.35** | **0.35** | 43.29 | 75.11 | 175.99 |

Table 3: Exploration study of betas on disentanglement (MIG) and reconstruction (Recon.). l1, l2, l3 denote [1,10,20,40,80], [1,10,40], [1,10] respectively. $s$ is fixed to 1.

Table 4: Comparison of scale. We report the mean$\pm$std of MIG and reconstruction for 5 trails on dSprites and Shapes3D. The sequence of betas is fixed to [1,10,40].

| Dataset | No. betas | MIG | Recon. | scale | MIG | Recon. |
|---------|-----------|-----|--------|-------|-----|--------|
| dSprites | l1 | 0.30$\pm$0.03 | 79.65$\pm$16.06 | 0.3 | 0.21$\pm$0.14 | **16.01$\pm$01.14** |
| | l2 | **0.35$\pm$0.02** | 51.99$\pm$26.99 | 0.5 | 0.29$\pm$0.09 | 22.40$\pm$01.78 |
| | l3 | 0.16$\pm$0.11 | **38.19$\pm$02.35** | 1.0 | **0.35$\pm$0.02** | 51.99$\pm$26.99 |
| Shapes3D | l1 | 0.54$\pm$0.06 | 65.01$\pm$25.37 | 0.3 | 0.55$\pm$0.02 | **24.43$\pm$01.37** |
| | l2 | **0.57$\pm$0.01** | 43.24$\pm$11.41 | 0.5 | **0.57$\pm$0.02** | 28.48$\pm$03.31 |
| | l3 | 0.55$\pm$0.04 | **39.31$\pm$06.96** | 1.0 | 0.57$\pm$0.01 | 43.24$\pm$11.41 |

way, the first layer aims to optimize the ELBO, and the subsequent layers optimize disentanglement jointly by DiT which works like a constraint of disentanglement. Therefore, the key of DeVAE is to connect the multiple latent spaces by DiT to form a hierarchical structure with a decremental IB.

**Effect of $\beta$.** More latent layers mean more chance to explore disentanglement solutions but need more time to converge. Though Wu. etc. (Wu et al., 2022) proposes the Annealing Test to determine the value of $\beta$, it requires labels to learn the information freezing point (IFP). Choosing a suitable $\beta$ for each layer is difficult without knowing the information of factors. Fortunately, DeVAE is insensitive to the choice of $\beta$, which means we can create redundant latent layers to cover all suitable $\beta$s. In Table 3, we compared tree cases: redundant betas ([1,10,20,40,80]), just betas ([1,10,40]), insufficient betas ([1,10]). Redundant betas slightly diminish the performance of disentanglement and reconstruction. It is an advantage to increase the parameter size and training iterations without rebooting

**Effect of Scale $s$.** Increasing $s$ will add the weights of higher beta, encouraging disentanglement more than reconstruction fidelity. It is a crucial hyperparameter to balance the objectives of latent layers. Note that our model equals the vanilla VAE when $s = 0$. In Table 4, we compared the effects of choosing $s$ and reported the mean$\pm$std scores of MIG (Chen et al., 2018) and reconstruction. For most cases, $s = 1$ is a good choice.

### 3.4 LIMITATION

Since our model creates several diverse latent spaces, it is a challenge to optimize multiple objectives. Though there are numerous combinations for setting pressures and weighting these objectives, we only search a limited range of hyper-parameters. Even so, DeVAE shows compatible performance on the benchmarks. Though we validated that our model is adequate for high-dimensional space, we did not test it on real problems. It is challenging to train a disentanglement model on large-scale problems, such as ImageNet.

## 4 RELATED WORK

**Disentanglement Learning.** Disentanglement learning aims at learning generative factors existing in the dataset, that is, disentangled representation learning. Though the definition of disentanglement is still an open topic (Kumar et al., 2018; Do & Tran, 2020; Abdi et al., 2019; Duan et al., 2020), it is widely accepted that the redundancy between latent variables diminishes disentanglement. Penalizing the Total Correlation (TC) (Watanabe, 1960) is an important direction in disentanglement learning, and many SOTA methods are based on it (Chen et al., 2018; Kim & Mnih, 2018; Esmaeili et al., 2019; Kumar et al., 2018; Wei et al., 2021). PM algorithm promotes factorial codes but only works for binary codes (Schmidhuber, 1992); Though ICA (Comon, 1994) and PCA (Wold et al., 1987) ensure independence theoretically, they extract linear representations. Until recently, deep learning has made it workable. FactorVAE (Kim & Mnih, 2018) applies an adversarial training method to approximate and penalize the TC term. $\beta$-TCVAE (Chen et al., 2018) decomposed the KL term into three parts: mutual information (MI), total correlation (TC), and dimensional-wise KL (DWKL). However, these methods rely on the estimation of TC, which is extremely hard for high-dimensional spaces.

**Information Bottleneck.** Information bottleneck theory (Tishby et al., 1999; Shannon, 1948) plays a vital role in interpreting neural networks. Some methods encourage disentanglement by increasing the information bottleneck while training (Jeong & Song, 2019; Burgess et al., 2018; Shao et al., 2022; Dupont, 2018; Wu et al., 2022). These methods vary in the way of expanding the IB. Cascade-VAE (Jeong & Song, 2019) sequentially relieves one latent variable at one stage to increase the IB. DynamicVAE (Shao et al., 2022) designs a non-linear PI controller for manipulating $\beta$ to control IB steadily increasing. DEFT (Wu et al., 2022) applies a multi-stage training strategy with separated encoders to extract one factor at one stage according to its information freezing point (IFP). However, the above incremental models, increasing the IB while training, suffer from the information diffusion (ID) problem (Wu et al., 2022) that the disentangled representation may diffuse the learned information into other variables. This work presents a novel framework with a decremental information bottleneck to solve the ID problem.

**Hierarchical Latent Variables.** Normalizing Flow (Rezende & Mohamed, 2015; Kingma et al., 2016) also uses hierarchical latent layers to generate an arbitrary distribution. Unlike Normalizing Flow, each layer aims to encourage disentanglement or reconstruction. Besides, Normalizing Flow gradually increases the complexity of the output distribution after entering a new layer. In contrast, our model reduces the complexity layer by layer. LadderVAE (Sønderby et al., 2016) also applies hierarchical latent variables in the encoder, but it using a symmetry structure decodes these latent variables in hierarchy. Therefore, the information among the $i$-th layer will increase comparing to the last layer.

## 5 CONCLUSION

We propose a novel framework with a decremental information bottleneck for disentanglement. Hierarchical latent spaces with disentanglement-invariant transformation are the key to overcoming the problem of losing disentanglement constraint while expanding the information bottleneck. The decremental method is compatible with high-dimensional problems and reduces the information diffusion problem. In practice, the typical disentanglement methods have to refine suitable hyperparameters for a dataset without labels by trail and error. In contrast, DeVAE is tolerant to redundant layers such that we can set a large parameter set to fit kinds of datasets.

**Broader Impact** Unlike previous works that spread the conflict of the trade-off over time, our work demonstrates a novel direction to solve the trade-off by spreading the conflict in spaces. Our work provides insights on balancing disentanglement and reconstruction.

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

Table 5: The architecture details. "FC." denotes fully connected layer, "conv." denotes convolutional layer, "deconv" denotes transposed convolution layer. $c$ is the dimension of color channel.

| Encoder | Decoder |
|---|---|
| $4 \times 4$ conv. 32 stride 2 | FC.256 |
| $4 \times 4$ conv. 32 stride 2 | FC. $4 \times 4 \times 64$ |
| $4 \times 4$ conv. 64 stride 2 | $4 \times 4$ deconv. 64 stride 2 |
| $4 \times 4$ conv. 64 stride 2 | $4 \times 4$ deconv. 32 stride 2 |
| FC. 256 | $4 \times 4$ deconv. 32 stride 2 |
| FC. 10 | $4 \times 4$ deconv. $c$ stride 2 |

# A APPENDIX

## A.1 ARCHITECTURE

The details of architectures are listed in Table 5.

## A.2 DISENTANGLEMENT-INVARIANT REPRESENTATIONS

In this section, we prove the proposed disentanglement-invariant transformation. Consider that we have a new representation by multiplying a diagonal matrix: $\boldsymbol{z}' = \boldsymbol{w}\boldsymbol{z}$, $\boldsymbol{w}$. We can calculate the Covariance between any two latent variables:

$$
\begin{aligned}
\text{Cov}(\boldsymbol{w}_i\boldsymbol{z}_i, \boldsymbol{w}_j\boldsymbol{z}_j) &= \mathbb{E}[(\boldsymbol{w}_i\boldsymbol{z}_i - \mathbb{E}[\boldsymbol{w}_i\boldsymbol{z}_i])(\boldsymbol{w}_j\boldsymbol{z}_j - \mathbb{E}[\boldsymbol{w}_j\boldsymbol{z}_j])] \\
&= \boldsymbol{w}_i\boldsymbol{w}_j(\mathbb{E}[\boldsymbol{z}_j] - \mathbb{E}[\boldsymbol{z}_i]\mathbb{E}[\boldsymbol{z}_j]) \\
&= \boldsymbol{w}_i\boldsymbol{w}_j\,\text{Cov}(\boldsymbol{z}_i, \boldsymbol{z}_j),
\end{aligned}
\tag{10}
$$

where the subscript denotes the index of latent variables. Note that we use a different notion in this section to simplify the formula.

Then we can get the correlation coefficient by

$$
\begin{aligned}
\rho(\boldsymbol{w}_i\boldsymbol{z}_i, \boldsymbol{w}_j\boldsymbol{z}_j) &= \frac{\text{Cov}(\boldsymbol{w}_i\boldsymbol{z}_i, \boldsymbol{w}_j\boldsymbol{z}_j)}{\sqrt{\text{Var}[\boldsymbol{w}_i\boldsymbol{z}_i]\,\text{Var}[\boldsymbol{w}_j\boldsymbol{z}_j]}} \\
&= \rho(\boldsymbol{z}_i, \boldsymbol{z}_j).
\end{aligned}
\tag{11}
$$

Therefore, the correlation matrix will not change by multiplying a diagonal matrix $w, w \neq 0$. We could create a disentanglement-invariant representation by multiplying a diagonal matrix.

## A.3 ESTIMATION OF $I(\boldsymbol{z}_j; \boldsymbol{c}_i)$

Given an inference network $q(\boldsymbol{z}|\boldsymbol{x})$, we use the Markov chain Monte Carlo (MCMC) method to get samples from $q(\boldsymbol{z})$ by the formula $q(\boldsymbol{z}) = q(\boldsymbol{z}|\boldsymbol{x})p(\boldsymbol{x})$. We use 10, 000 points to estimate $q(\boldsymbol{z})$. Then, we discretize these latent variables by a histogram with 20 bins. After discretizing one latent variable, we call a discrete mutual information estimation algorithm to calculate $I(\boldsymbol{w}_j\boldsymbol{z}_j; \boldsymbol{c}_i)$ by a 2D histogram.

## A.4 LATENT TRAVERSALS

We compare DeVAE to others with latent traversals on Shapes3D and dSprites. Each column denotes the generated images by traversing one latent variable from -2 to 2. We also interpret the extracted factor at the bottom. From Figure 6 and Figure 7, we can see that DeVAE has a lower entanglement level. Note that only DeVAE disentangles object size isolated on Shapes3D.

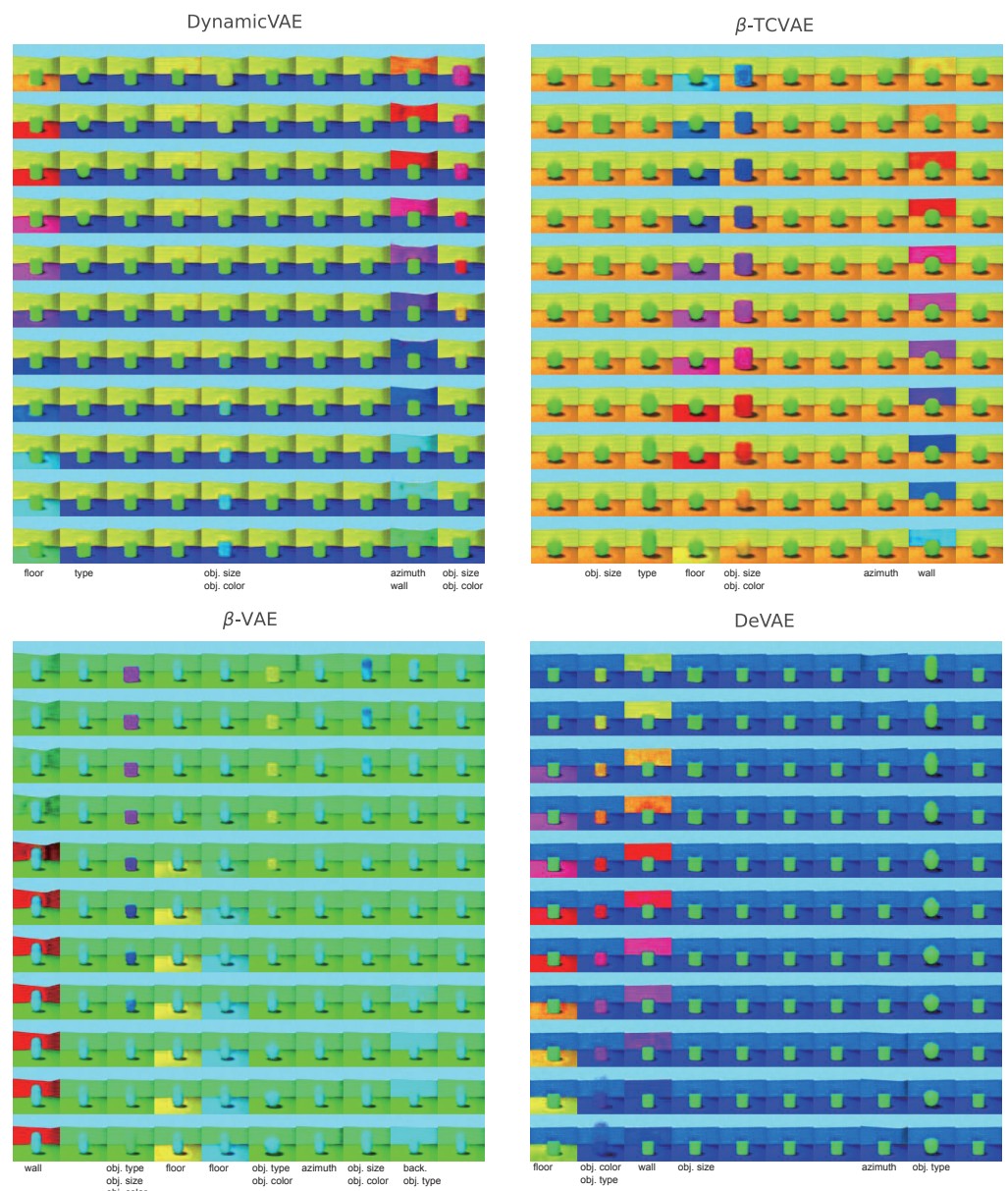

Figure 6: Latent traversal on Shapes3D. "back." denotes background color, "floor" denotes floor color, "obj." denotes object, and "wall" denotes wall color.

## A.5 RANDOM SAMPLING

We show the visualization of random sampling from the best models with the highest MIG trained on dSprites and Shapes3D in Figure 8 9 10.

## A.6 DECREMENTAL DIAGRAM

Figure 11 shows how the mutual information between factors and latent variables decreases over layers on dSprites. One can see the mutual information decreases along the layers, and information of shape, scale, and orientation is totally disappeared in layer2. The last layer is more likely to

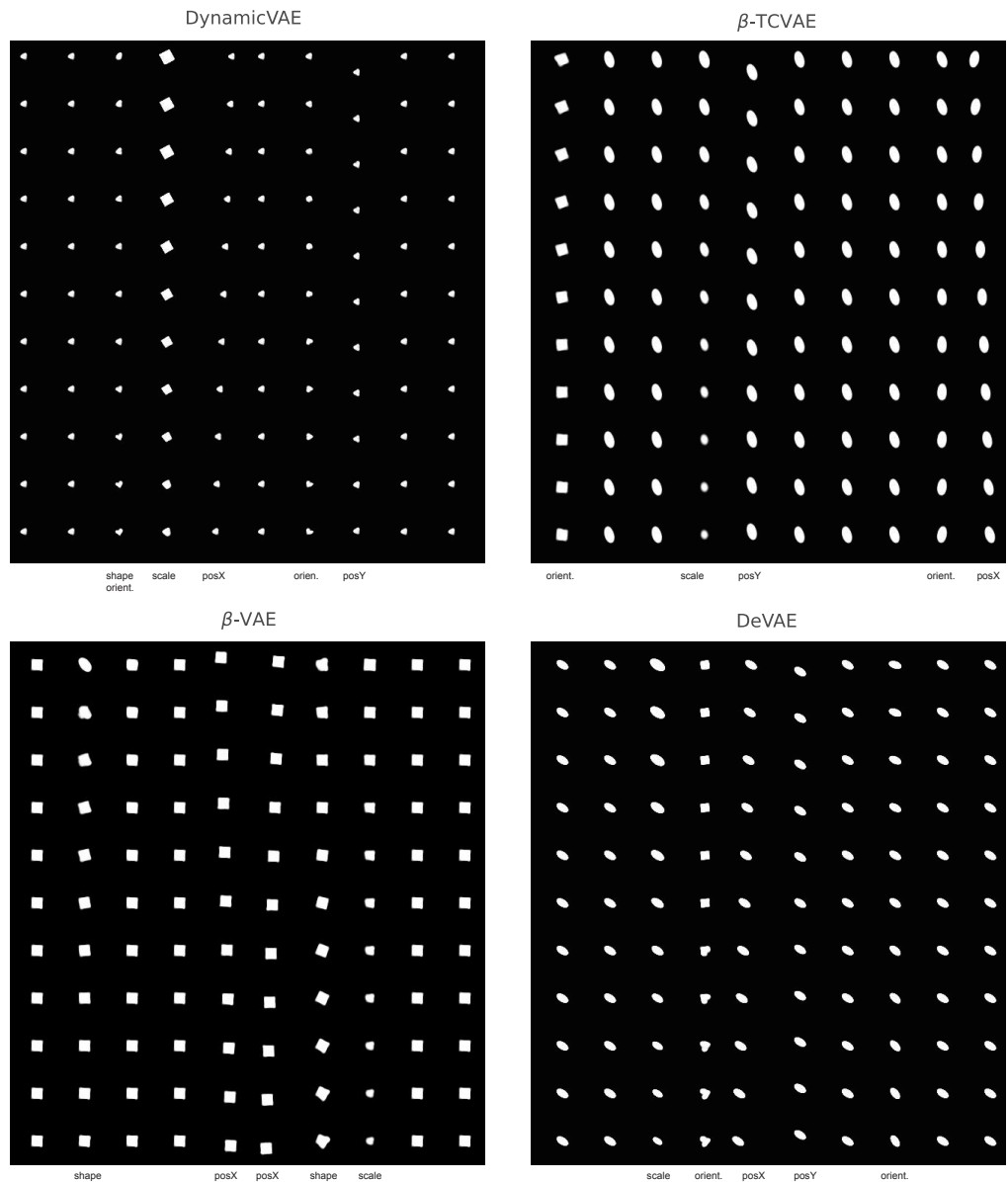

Figure 7: Latent traversal on dSprites.

disentangle them, and that property will be preserved and passed to the first layer for a constraint of disentanglement.

## A.7 HIGH-DIMENSIONAL LATENT SAPCE

DeVAE has significant advantages for handling high-dimensional latent spaces. Though Dynamic-VAE outperforms low-dimensional latent spaces, there is a gap in the high-dimensional latent spaces. We trained DynamicVAE and DeVAE with 1024-dimensional latent spaces on dSprites to investigate what causes the difference. (Cao et al., 2022) found that existing disentanglement metrics fail to make meaningful measurements for high-dimensional representation models, therefore we apply the proposed metric by them in this experiment. Active variables denote the latent variables containing

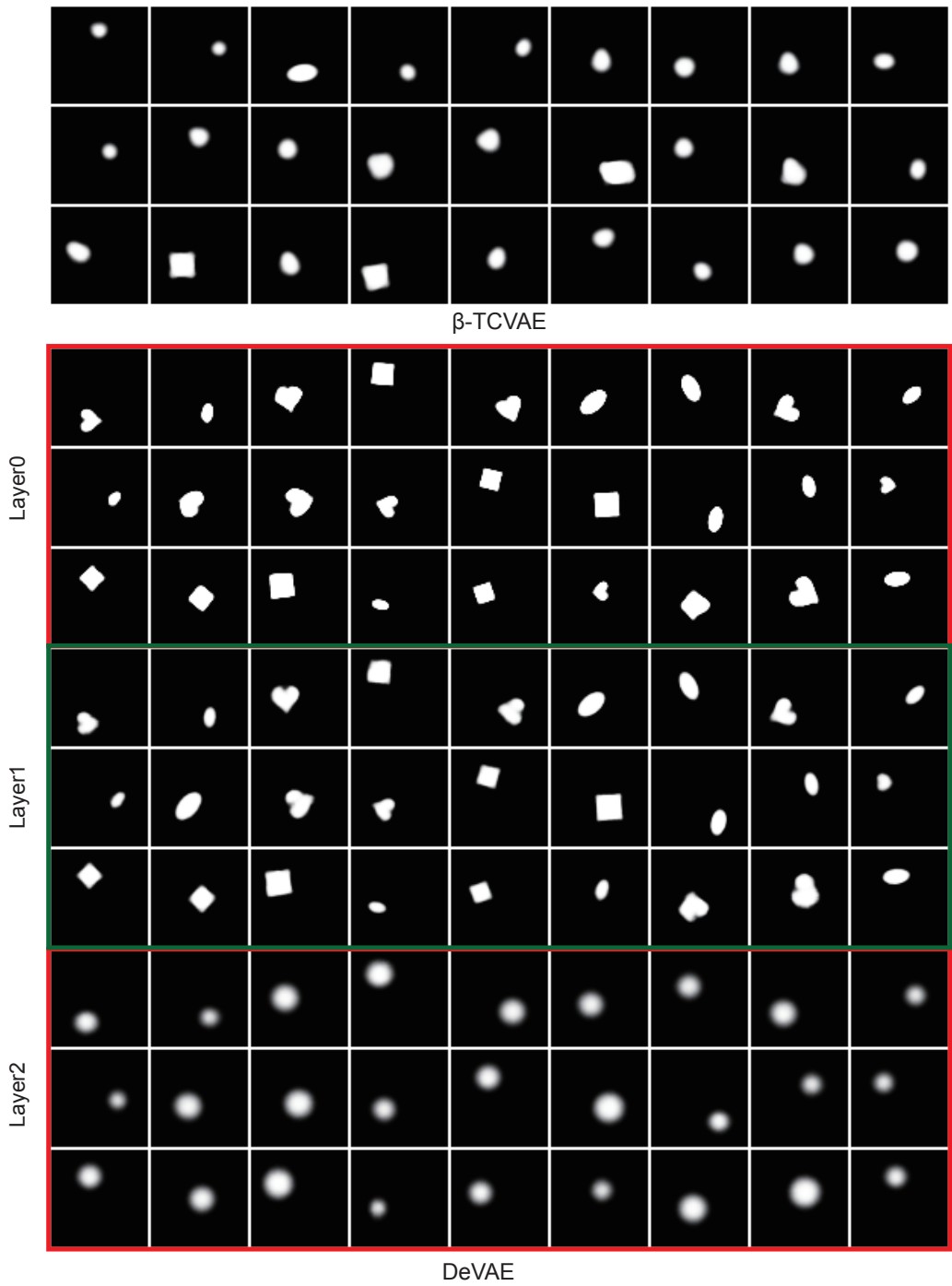

Figure 8: Generated images of $\beta$-TCVAE and DeVAE from sampling random noise on dSprites.

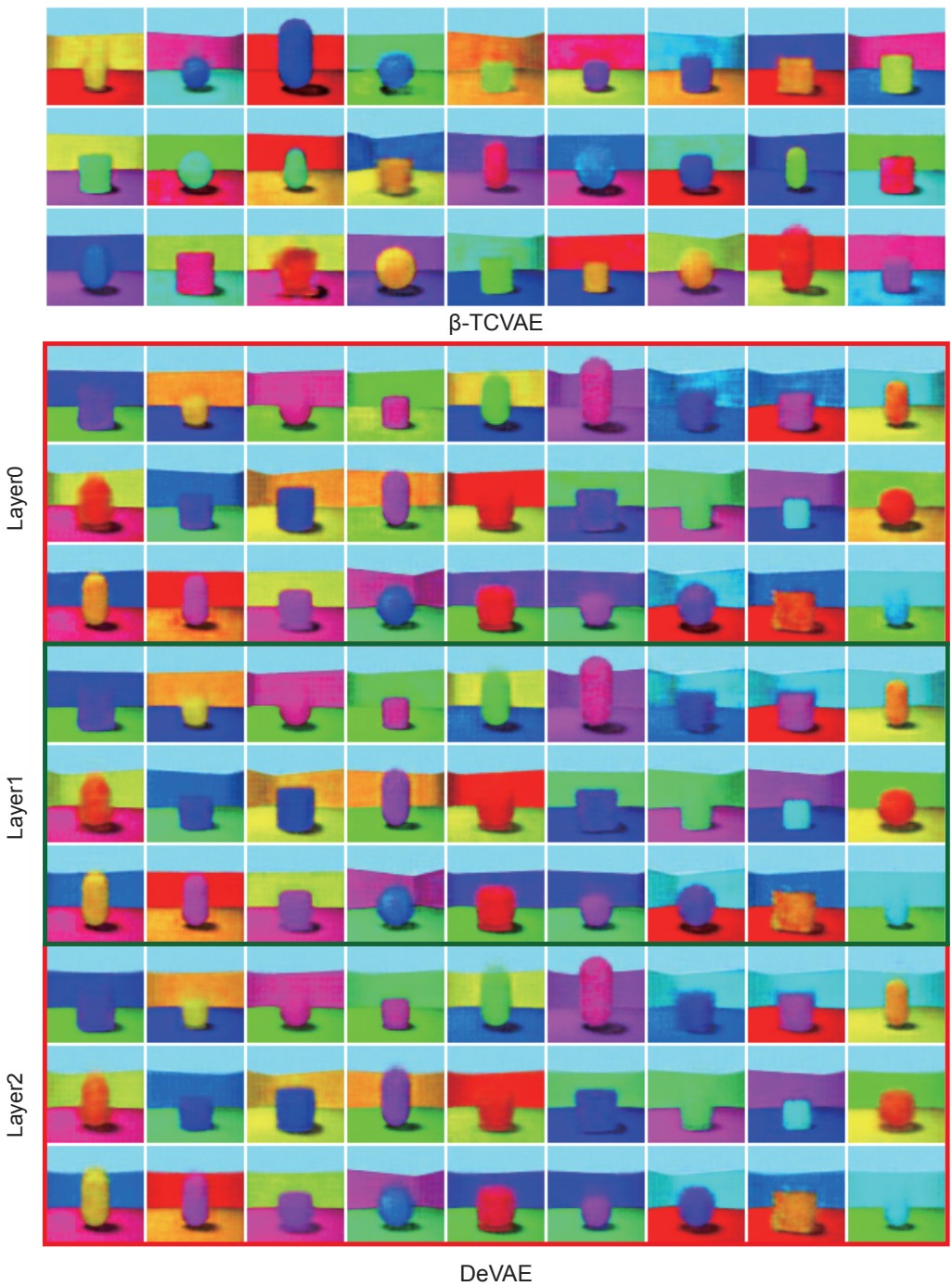

Figure 9: Generated images of $\beta$-TCVAE and DeVAE from sampling random noise on Shapes3D.

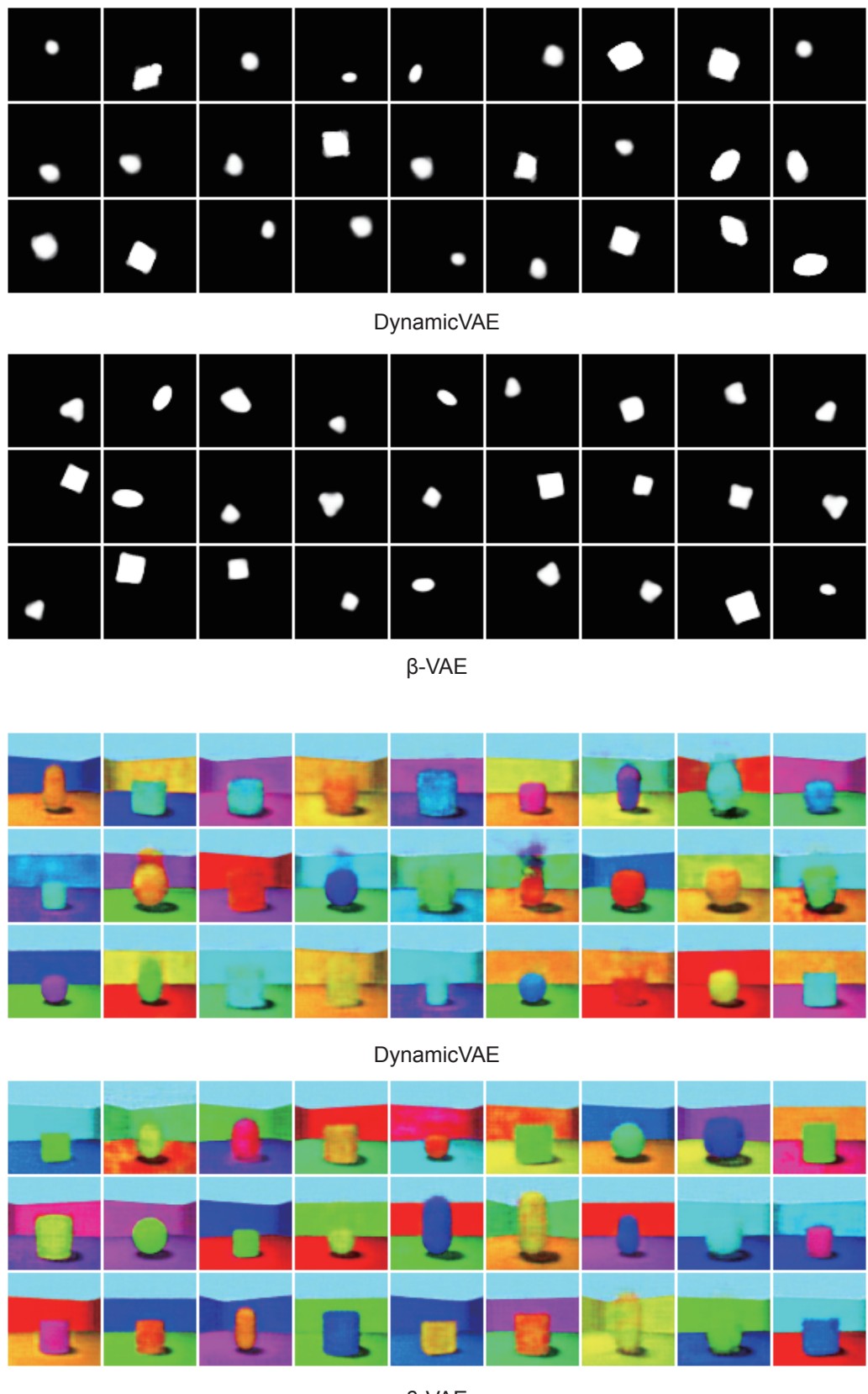

Figure 10: Generated images of DynamicVAE and $\beta$-VAE from sampling random noise on dSprites and Shapes3D.

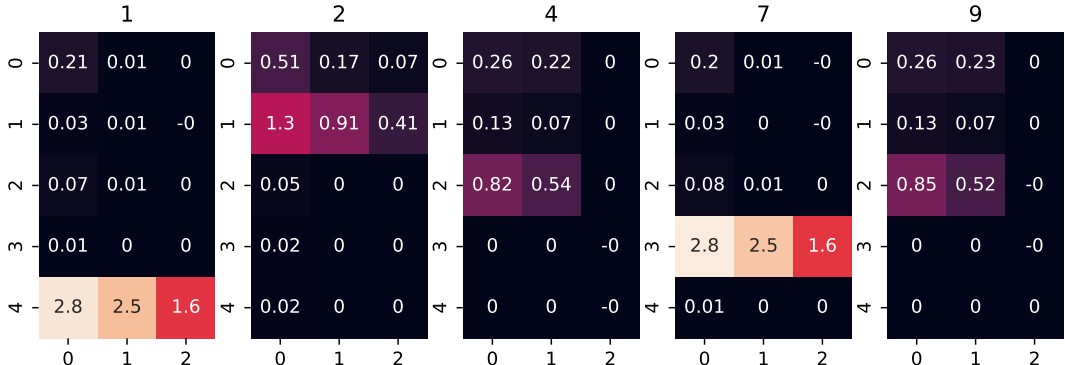

Figure 11: The mutual information between factors and latent variables over layers on dSprites. We select five informative variables, and the title denotes the index of the latent variable. The rows denote the factors, shape, scale, orientation, posX, and posY respectively. The column denotes the layer where the latent variable (title) is.

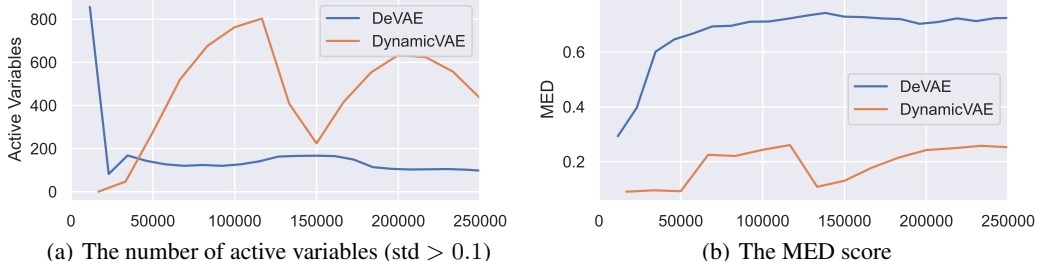

(a) The number of active variables (std $> 0.1$)

(b) The MED score

Figure 12: Comparison of active variables and MED scores on 1024-dimensional models on dSprites. DynamicVAE is unstable while expanding the information bottleneck.

Table 6: Comparison of reconstruction error (Recon.), MIG score, and ELBO for six disentanglement methods.

| dataset | model | Recon. | MIG | ELBO |
|---------|-------|--------|-----|------|
| dSprites | FactorVAE | 21.55±0.84 | 0.34±0.04 | -46.05±2.24 |
| | CascadeVAE | 12.04±1.23 | 0.20±0.07 | -32.14 ± 1.29 |
| | Dynamic | 19.81±1.19 | 0.35±0.01 | -37.83±1.17 |
| | beta-TCVAE | 73.04±3.41 | 0.29±0.10 | -82.29±3.71 |
| | beta-VAE | 48.75±2.84 | 0.17±0.05 | -61.17±3.13 |
| | DeVAE | 36.02±20.02 | 0.32±0.11 | -51±22.26 |
| Shapes3D | FactorVAE | 18.48±2.28 | 0.38±0.28 | -38.08±1.87 |
| | CascadeVAE | 14.84±1.98 | 0.46±0.11 | -32.54±2.10 |
| | Dynamic | 29.70±4.15 | 0.54±0.04 | -47.68±4.28 |
| | beta-TCVAE | 44.53±5.69 | 0.49±0.11 | -60.01±6.29 |
| | beta-VAE | 34.95±2.34 | 0.42±0.18 | -49.09±2.72 |
| | DeVAE | 46.80±13.97 | 0.52±0.10 | -74.73±31.66 |

information, and those containing no information will collapse into one point, so the active variables will have large variances. From Figure 12, DynamicVAE has more active variables and performs worse than DeVAE. DynamicVAE expanding the IB smoothly could have a good performance on low-dimensional spaces, but the increment of dimensions raises the chance of leaking information to others. As a result, the number of active variables increases quickly when expanding the IB, see iteration 20000 to 100000.

## A.8 COMPARISON WITH FACTORVAE AND CASCADEVAE

FactorVAE (Kim & Mnih, 2018) and CascadeVAE (Jeong & Song, 2019) are two relevant methods for disentanglement. We compare six disentanglement methods on dSprites in Table 6. Similar to $\beta$-TCVAE, FactorVAE can not consistently outperform one hyperparameter on two datasets. Though CascadeVAE has good reconstruction fidelity, it cannot disentangle all factors properly.

