# OpenReview forum: "Variational Autoencoders with Decremental Information Bottleneck for Disentanglement"
_ICLR.cc/2023/Conference — Submitted to ICLR 2023_

### Official Review · Reviewer_Z9wf · 2022-10-17

**Confidence:** 4
**Correctness:** 2
**Technical Novelty And Significance:** 3
**Empirical Novelty And Significance:** 2
**Recommendation:** 3

**Clarity, Quality, Novelty And Reproducibility:**

Novelty: The idea is, to the best of my knowledge, novel and it is interesting. Using different latent spaces in parallel to address both reconstruction and representations is a general strategy that could be viable for addressing this very real problem in representation learning/generative models.

Clarity: The method is not clear in my opinion. In particular, the fact that variables are shown in the main architecture figure (fig 1), but never mentioned or introduced in equations is a major clarity flaw. Some mechanisms about the decoding are also just explain in-text and are difficult to follow.

Quality: The quality is a bit lacking in the experiments. Most experiments do not clearly show the important features claimed in the abstract intro and technical section, i.e., the better tradeoff between reconstruction and disentanglement. The lack of axis in a lot of the graphs is also a major quality flaw.

Reproducibility: The experiments are reproducible, aside of the confusions about the architecture mentioned in the quality section.

**Strength And Weaknesses:**

Strengths:

The idea to have parallel latent spaces with different betas and using them all to obtain "the best of both worlds" is interesting and, to the best of my knowledge, sufficiently novel in this form.


Weaknesses:

I have several concerns regarding the technical description and experiments.

Technical Section:
- The description lacks clarity in some places. For example, it is not clear what is the role of the vectors(?) v_i shown in figure 1, as they are never mentioned in the text and do not appear in the ELBO of equation 2 or the transformation tau_i of equations 3-6. Therefore, I could not understand how the decoder works; does it individually decode each z_i with the same weights and add up the reconstruction costs?
- According to theorem 1, the disentanglement of z_0 is preserved in all subsequent latent spaces. This is inconsistent with the aim of the method, where, as phrased in the paper "the sequential layers aim to disentangle factors of variation by setting narrow bottlenecks". The representations' disentanglement cannot be changed by scaling. I may have got this wrong somewhere, but if this is the case, this is a critical flaw.

Experiments:
The experimental section is rich, but in most cases hard to interpret. My main concerns are the following:
-  The results of figure 2 have several issues (at least in their presentation): 1) the plots show 3 disentanglement metrics and 1 reconstruction metrics. As per the objective of the method, what counts is their tradeoff, so a much better way to show performance would be to do something more similar to figure 5 and plot each of the 3 disentanglement metrics vs. reconstruction error. 2) It is not clear where better is higher and where better is lower. Clearly lower rec. error is better, but do we want high or low MIG for example? From the descriptions below it is not clear. 3) In the text it is stated that the DeVAE is the best, but in this plots it is often somewhere in the middle, so it is not clear how the reader is meant to interpret these plots as proving superior performance of the proposed method.
- Figure 3 is not interpretable at all. I think the columns (x-axis) are the 3 random samplings, but what is the 2-6 numbers above? Where is the -2 to +2 positions? what is the y-axis? There is no comparison with any other method, so the "disentangled perfectly" mentioned in the text is not very meaningful.
- Figure 4 has a similar problem to figure 2; The information diffusion shows how well the model is disentangling, but we need also to see the reconstruction. As stated also in this paper, it is their relationship that counts.
- Figure 5 shows, in my opinion, the relevant information; reconstruction vs. a measure of disentanglement. However, these results raise some doubts. Assuming MIG is better when higher, for high dimensions the proposed method is the best, but for low dimensions, it is outperformed significantly in both MIG and reconstruction error by the DynamicVAE. Is this right?


**Summary Of The Paper:**

The paper presents a novel hierarchical latent space structure and regularisation strategy (beta-pressure) to improve the balance of representation and information retention, or reconstruction quality, in VAEs. The latent space is built to have a Markov chain-like structure, where, in the encoder, the first latent space is conditioned on inputs q(z_0|x), just like standard VAEs, and all subsequent encodings are conditioned on the previous one, i.e., q(z_i+1|q_z_i). The decoder reconstructs the input from each latent space in parallel. Although I am not entirely clear on the whole mechanism (see below).

**Summary Of The Review:**

The novelty of the paper is sufficient in my opinion, but the clarity and quality do not meet the bar. The technical section is not sufficiently clear and the experiments are difficult to read and not sufficiently conclusive.

---

> ### Author Response · Authors · 2022-11-15
> **Language improvement**
>
> We have rewritten Section 2 to improve the clarity in this revision.
>
> - the role of the vectors(?) $v_i$ shown in figure 1
>
> ${v}_i\in\mathbb{R}^{1\times D}$ denotes a learnable layer embedding for the $i$th layer to identify the layer index for the coder, like PosEmbed in ViT.
>
> - the transformation $\tau_i$ of equations 3-6
>
> The transformation $\tau_i$ is explained in Section 2.3. Actually, there are plenty of implementations of $\tau_i$, and we only discuss a simple one to gain disentanglement-invariant, see Equation 4.
>
> - does it individually decode each $z_i$ with the same weights and add up the reconstruction costs?
>
> You are right that one decoder network independently takes $z_i$ concatenating a layer embedding $v_i$, but DeVAE does not add up all reconstructions, instead, it only selects one layer to optimize in a mini-batch.
>
> - Figure 3 is not interpretable at all.
>
> We update Figure 3 and the caption.
> Latent traversal of DeVAE with the best MIG score on dSprites. Each block shows the
> generated images of traversing the latent variable from −2 to 2. Each group shows the traversal
> images sampling from 3 different random noise. The title above each group denotes the index of
> latent variable.
> Due to the limitation of paper length, we put the comparison to the appendix, see Figure 6-9.
> One can see that only DynamicVAE and DeVAE generate clear shapes for the "heart" in Figure 7.

---

> ### Author Response · Authors · 2022-11-16
> **Response to Reviewer Z9wf**
>
> We appreciate your review and suggestions. Below are our responses to your questions.
>
> Q1: This is inconsistent with the aim of the method, where, as phrased in the paper "the sequential layers aim to disentangle factors of variation by setting narrow bottlenecks".
>
> A1: The disentanglement of all layers is the same, which is determined by all objectives not only $z_0$.
> Ideally, the first layer will be promoted to disentanglement when the subsequent layers are disentangled.
> The disentanglement-invariant transformations are still learnable to adjust the representations for the objectives in layers, therefore the model can optimize different objectives in separated layers, i.e., the first layer aims for reconstruction, and the last layer aims for disentanglement.
> We update Section 2.1 to better explain the objectives.
>
> Q2: Clearly lower rec. error is better, but do we want high or low MIG for example?
>
> A2: For all disentanglement metrics, a higher value means better disentanglement.
>
> Q3: how the reader is meant to interpret these plots as proving superior performance of the proposed method.
>
> A3: "best" is inaccurate and overstated, and we modify the description in Section 3.2.
> It refers to the robustness of hyperparameters and model size for achieving a balance without significant shortages.
> Though some works report SOTA results, their performances highly depend on the selection of hyperparameters.
> In practice, refining the proper hyperparameter is extremely hard to learn a disentangled model on a dataset without labels.
>
> Q4: Figure 4 has a similar problem to figure 2; The information diffusion shows how well the model is disentangling, but we need also to see the reconstruction.
>
> A4: We show the reconstruction of the best models in Figure 8-10.
> But the ID problem can not be revealed from the reconstruction.
> The ID problems mean that the information of one factor leaks into other variables, but the overall information of the latent space can be unchanged or increased.
> Therefore, the reconstruction error can decrease even if the ID happens.
>
> Q5: for low dimensions, it is outperformed significantly in both MIG and reconstruction error by the DynamicVAE.
>
> A5: DeVAE is the best model for high-dimensional latent spaces, but it does not outperform significantly for low dimensions. Though they show distinct differences in these disentanglement metrics, it is hard to see the advantages of the best models from Figure 6 7. These metrics are not gold rules like accuracy, because the MIG can achieve 0.4 ideally if the model only extracts posX and posY, and drops other factors.  That is the case DeVAE does in Layer2, see the sampling images from Figure 8 9.
> We investigate why \btcvae{} and DynamicVAE have significant performance decay in Figure 5.
> Higher dimensional space increases the complexity of calculating the TC and leads to significant estimation errors and also increases the chance of the ID problem for DynamicVAE see Appendix A.7.
> \betavae{} and DeVAE show robustness in high-dimensional latent spaces, which is necessary to train a large model on large data.
> We update the explanation in Section 3.2 Scaling to High-dimensional Latent Space.

---

### Official Review · Reviewer_nZJo · 2022-10-24

**Confidence:** 4
**Correctness:** 2
**Technical Novelty And Significance:** 2
**Empirical Novelty And Significance:** 2
**Recommendation:** 3

**Clarity, Quality, Novelty And Reproducibility:**

**Clarity**

The method part of the paper is unclear while the experiment part is  overall clear. Please see the weakness part.

**Novelty**

The proposed method has some overlaps in previous methods that propose hierarchical latent variables, such as ladderVAE, but is novel in using the diagonal transformation matrix.

**Quality**

The paper provides reasonable empirical studies but lacks theoretical analysis.

**Reproducibility**

The authors include details of the algorithm and model parameterization. Although the method formulation is ambiguous, it should be reproducible after the clarification.

**Strength And Weaknesses:**

**Strength**

The proposed method is not complex and easy to understand. The experiments are comprehensive and sufficient ablation studies are provided to support the claims.

**Weakness**

The method part is not properly stated; some mathematical expressions might be wrong (at least they cannot correspond to either Fig 1. or Alg. 1). I am confused after reading it:

- According to Fig 1. and Alg. 1, the latent variables depend on previous ones, i.e., $z_1$ is dependent to $z_0$, $z_2$ depends on $z_1$. However, Equation 2 seems to show $z_i$ is only dependent to $x$ or $z_0$, and $z_1, z_2, ..., z_K$ are independent like K VAEs. Equation 3, also seems to have $z_i$ marginalized out, while the code in Alg. 1 does not have such an operation. The authors need to clarify this part.

- The notation v is in Fig. 1 is not used in the mathematical definition.

- What is j in Equation 5-6?

The paper lacks theoretical analysis. The choice of $\beta$ does not have clear guidance. The authors do not show whether the modified ELBO is still valid and whether it is a tight bound.

According to the results in Table 2, by adding HiS and DiT, the reconstruction fidelity seems to get worse, which does not support the claim of the paper: "optimize reconstruction and keep disentangled representation".

In the case when the latent space is stacked with redundant variables, a non-trivial study could be whether there are some variables that are more important while some variables can be distilled.

The granularity of the disentanglement of the proposed method is correlated to the parameter size and training efficiency. In other words, if we would like to refine the granularity of the information bottleneck (have more choices of $\beta_i$ like done in Table 3), the parameter size and required training iterations also increase.

From the perspective of practical use, the improvement is a little bit marginal, and the experiments are conducted on small-scale data. It is still unclear whether this method could be useful for larger data and models.

**Minors**

Equation 4: some 'i's are bold while some are not.
Equation 5: same issue.

Below Equation 7: $p\left(\boldsymbol{z}_i\right)=(1-s) s^i /\left(1-s^{i+1}\right), s \neq 1 ; \frac{1}{K}, s=1$ is not a rigorous math expression.

Section 4. Paragraph1-Line7: one citation is missing.

**Summary Of The Paper:**

The paper presents a hierarchy of latent representations weighted by a series of monotonically increasing hyper-parameters, which compose an information bottleneck. The authors empirically demonstrate their idea on dSprites and Shapes3D to show the proposed method is able to learn disentangled representations while preserving reconstruction fidelity.

**Summary Of The Review:**

The disentanglement of representation has been an important topic in the study of representation learning and has been studied for a long time along the development of information bottleneck and variational autoencoders. The proposed method has its contributions in improving the disentanglement of the latent variables while preserving the reconstruction quality. Both the quality and clarity of the paper still need improvement. I am inclined to consider this paper below the acceptance threshold given the current version.

---

> ### Author Response · Authors · 2022-11-15
> **Response to Reviewer nZJo**
>
> We appreciate your review and suggestions. Below are our responses to your questions.
>
> Q1: The paper lacks theoretical analysis. The choice does not have clear guidance. The authors do not show whether the modified ELBO is still valid and whether it is a tight bound.
>
> A1:
> The first layer is still a valid ELBO, but it is affected by the subsequent layers to add the constraint of disentanglement, which will get a worse ELBO.
> Similarly, the L2 regularization will hurt the training error but gain better generalization.
> It is a limitation to be unable to give strict proof, but we can still empirically compare the ELBO from the following table.
>
> |  dataset |    model   |      Recon.     |      MIG      |        ELBO       |
> |:--------:|:----------:|:---------------:|:-------------:|:-----------------:|
> | dSprites |  FactorVAE |  21.55±0.84 | 0.34±0.04 |  -46.05±2.24  |
> |          | CascadeVAE |  12.04±1.23 | 0.20±0.07 | -32.14 ± 1.29 |
> |          |   Dynamic  |  19.81±1.19 | 0.35±0.01 |  -37.83±1.17  |
> |          | beta-TCVAE |  73.04±3.41 | 0.29±0.10 |  -82.29±3.71  |
> |          |  beta-VAE  |  48.75±2.84 | 0.17±0.05 |  -61.17±3.13  |
> |          |    DeVAE   | 36.02±20.02 | 0.32±0.11 |   -51±22.26   |
> | Shapes3D |  FactorVAE |  18.48±2.28 | 0.38±0.28 |  -38.08±1.87  |
> |          | CascadeVAE |  14.84±1.98 | 0.46±0.11 |  -32.54±2.10  |
> |          |   Dynamic  |  29.70±4.15 | 0.54±0.04 |  -47.68±4.28  |
> |          | beta-TCVAE |  44.53±5.69 | 0.49±0.11 |  -60.01±6.29  |
> |          |  beta-VAE  |  34.95±2.34 | 0.42±0.18 |  -49.09±2.72  |
> |          |    DeVAE   | 46.80±13.97 | 0.52±0.10 |  -74.73±31.66 |
>
> Though the ELBO of DeVAE is relatively low and has a large variance,  DeVAE has the potential to get an ideal disentangled model. We visualize the best models with the highest MIG in Appendix A.4 and A.5. Compared with other methods, the reconstructed images of DeVAE are clear, and we can see the shape 'heart'  from Figure 8. Meanwhile, most dimensions represent one factor from Figure 7.
>
> Q2: According to the results in Table 2, by adding HiS and DiT, ...
>
> A2: DeVAE does keep disentanglement unchanged by adding HiS and DiT, meanwhile, the reconstruction fidelity is becoming better and better from layer 2 to layer 0 in Table 2.
> Our claim means that the reconstruction fidelity is relevant high when keeping a high disentanglement performance.
> We agree that the reconstruction fidelity has a slight degradation for the effects from the subsequent layers, but we argue that HiS with DiT introduces a disentanglement constraint into the VAE and achieves a balance between disentanglement and reconstruction fidelity.
> We update the description of DeVAE  Section 1 and 2.2.
>
> Q3: In the case when the latent space is stacked with redundant variables, a non-trivial study could be whether there are some variables that are more important while some variables can be distilled.
>
> A3: Thank you for your advice.
> We supplement the visualization of information changes over layers in Appendix A.6. From Figure 11, we can see that Layer0 captures all factors; Layer1 filters shape on latent variable 2; Layer2 filter shape and orientation.
>
> Q4: The granularity of the disentanglement of the proposed method is correlated to the parameter size and training efficiency.
>
> A4:  It is an advantage for DeVAE to increase the parameter size and training iterations without rebooting.
> The previous study shows that the disentanglement performance of the popular disentanglement methods is highly dependent on the selection of hyperparameters.
> To get a good model, these methods need lots of trial and error.
> What's worse, the datasets from reality have no label to refine these models.
> In contrast, DeVAE can use a large parameter size with redundant betas to cover the suitable hyperparameters.
>
> Q5: It is still unclear whether this method could be useful for larger data and models.
>
> A5: This work does not aim to prove DeVAE can work for large data. Our results show that the Hierarchical Latent Spaces with Disentanglement-invariant Transformation is compatible with large models, which is a necessary condition to apply to large data.
> The real dataset, like ImageNet, is too complex to get the ground-truth factors to evaluate the disentanglement.
> It's unlikely to access disentanglement methods on large data directly.
> In this phase, we still need the artificial dataset to validate the workability of the proposed method.
> We find a work using a similar setting to explore disentanglement of the self-supervised learning on high-dimensional latent spaces [1].
>
>
> Reference:
>
> 1. Jinkun Cao, Ruiqian Nai, Qing Yang, Jialei Huang, and Yang Gao. An empirical study on disentanglement of negative-free contrastive learning. In Alice H. Oh, Alekh Agarwal, Danielle Belgrave, and Kyunghyun Cho (eds.), Advances in Neural Information Processing Systems, 2022. URL https://openreview.net/forum?id=fJguu0okUY1.

---

> > ### Comment · Reviewer_nZJo · 2022-12-03
> > **Response to the authors**
> >
> > I appreciate the authors' efforts in addressing all the comments. After reading the revision and the responses, I found the quality of the paper is indeed improved. Unfortunately, I still have concerns regarding the method part. The latest formulation of Eq (2) involves variable $v_i$, while it is not clear how this variable is learned and from which distribution it is sampled. So far it is hard to say whether this is a valid ELBO as claimed by the authors. Moreover, only adding $v_i$ as a condition of the distribution does not distinguish the proposed method from $\beta$-VAE. I decide to keep my rating unless we can see significant improvement or clarification about this perspective.

---

> ### Author Response · Authors · 2022-11-15
> **Language Improvement**
>
> We have updated the method part in Section 2 and added details in Figure 1.
>
> -  According to Fig 1. and Alg. 1, the latent variables depend on previous ones.
>
>
> The latent variables $z_0, z_1, z_2, ..., z_{K-1}$ are sampled independently, but their parameters $\mu_1,...,\mu_{K-1}$ are dependent on previous ones through the proposed disentanglement-invariant transformation.
>
> - The notation v in Fig. 1 is not used in the mathematical definition
>
> ${v}_i\in\mathbb{R}^{1\times D}$ denotes a learnable layer embedding for the $i$th layer to identify the layer index for the coder, like PosEmbed in ViT.
>
> - What is j in Equation 5-6
>
> The upper j in Equation 5-6 is a mistake, which should be i.
> The lower j in Equation 5-6 iterates layers from layer 0 to i-1 to calculate the parameters of the latent variables in the $i$-th layer by a  bottom-up process.
> We have updated Equation 2-4 and Figure 1 in the revision.

---

### Official Review · Reviewer_8f7N · 2022-10-28

**Confidence:** 4
**Correctness:** 2
**Technical Novelty And Significance:** 3
**Empirical Novelty And Significance:** 2
**Recommendation:** 5

**Clarity, Quality, Novelty And Reproducibility:**

Clarity - The paper is not very clearly written and is difficult to understand at various instances. Most of the design decisions are not supported with justifications, some of them are mentioned in weaknesses above.

Quality - The proposed method is technically sound but the quality of experiments is lower than what I would expect from an ICLR paper.

Novelty - The contributions of the paper seem to be novel.

Reproducibility - The paper provides many experimental details and a pseudocode, although many design choices are unexplained and/or maybe based on test results instead of on a validation set.

**Strength And Weaknesses:**

Strengths
- The paper focuses on the balance between disentanglement and reconstruction fidelity, an important problem in the field of disentangled representation learning.
- The proposed method is technically sound.
- The proposed method performs better than beta-VAE and beta-TCVAE on dSprites and shapes3D datasets.
- Ablation studies show the importance of each proposed component and selection of hyperparameters.


Weaknesses:

- The paper has low readability. A lot of the issues are certainly fixable but in its current form, it is confusing enough to distract from evaluating the technical contributions of the paper. Certain examples are:
    - “DeVAE surpasses 2% for β-TCVAE and 9% for β-VAE.” In what terms exactly?
    - Many instances in the introduction talking about spreading the conflict of disentanglement and reconstruction over time and space is not easy to follow and understand.
    - “However, in this work, we get rid of calculating TC by leveraging the narrow information bottleneck (Tishby et al., 1999; Burgess et al., 2018) to find efficient codes for representing the data, which promotes disentanglement.” This is pretty confusing right where it is in the introduction and only becomes somewhat clear after reading the method section.
- The authors claim that DynamicVAE suffers from Information Diffusion problems. If that is the case, wouldn’t that result in low disentanglement scores or at least high variance across different seeds for DynamicVAE? But, that’s not the case in Figure 2.
- The experiment for high-dimensional latent space is weak. 1024 dimensional latent space for dSprites seems unrealistic, it ideally should be for a dataset that requires high-dimensional latent space. And, there are no quantitative numbers. Why is DeVAE worse for low dimensions? A more high-level question would here is — what exactly in DeVAE makes it more compatible for handling high-dimensional latent space?
- A lot of design decisions are unexplained. How are the hyperparameters for other methods chosen? Are the beta values chosen for DeVAE selected based on test performance or on a validation set?
- What are the layer embeddings? They seem to be an important component of the method but are never explained.
- How much is the computational overhead because of the hierarchical latent space and how does it compare to other methods that are compared within the paper?
- Why is DeVAE not compared with FactorVAE and Cascade-VAE, they both seem highly relevant as well.

**Summary Of The Paper:**

The paper proposes a new method called DeVAE for learning disentangled representations based on beta-VAE. Specifically, the proposed method builds a hierarchical latent space with disentanglement-invariant transformation between them and decreases information bottleneck layer-by-layer to balance disentanglement and reconstruction fidelity. Experimental results show that DeVAE performs comparably as previous works. DeVAE seems to perform better than other methods when the latent dimension is high.

**Summary Of The Review:**

The paper focuses on the important problem of the balance between disentanglement and reconstruction fidelity in disentangled representation learning research. The proposed method is technically sound. However, there are significant concerns with the experiments in the paper and the paper clarity is low. Hence, I would recommend the paper for rejection in its current form.

---

> ### Author Response · Authors · 2022-11-15
> **Response to Reviewer 8f7N**
>
> We appreciate your review and suggestions. Below are our responses to your questions.
>
> Q1: DynamicVAE suffers from Information Diffusion problems
>
> A1: From Figure 4, we can see that the absolute value of decreased information is low (from about 0.3 to 0.05 for shape).
> Therefore, that won't result in a large decrement in disentanglement scores.
> We can see that another IB-based method CascadeVAE has lower disentanglement scores from Table 6 in the revision or below this response.
> The change of IB is smooth for the PID controller applied by DynamicVAE, which relieves the ID problem.
> We investigate why DynamicVAE is worse for high dimensions in Appendix A.7.
> The size of the latent space magnifies the chance of leaking information.
> Therefore, DynamicVAE has a significant performance gap between low-dimensional space and high-dimensional space.
>
> Q2: The experiment for high-dimensional latent space is weak. 1024 dimensional latent space for dSprites seems unrealistic, it ideally should be for a dataset that requires high-dimensional latent space. And, there are no quantitative numbers.
> Why is DeVAE worse for low dimensions?
> A more high-level question would here is — what exactly in DeVAE makes it more compatible for handling high-dimensional latent space?
>
> A2:
> It is a pity that we have to evaluate models on an artificial dataset with few factors because It is almost impossible to get the ground-truth factors of the images from real life. We argue that working on high-dimensional latent space is a necessary condition to apply disentanglement methods to real problems, and some works start to explore this [1].
> We can see that DeVAE is more compatible for handling high-dimensional latent space from Appendix A.7.
>
> Q3: How much is the computational overhead?
>
> A3: The increased computation comes from expanding the last FC layer of the encoder and introducing layer embeddings on the decoder, which equals adding two linear layers.
> Though DeVAE has several latent spaces, we randomly choose and optimize one layer in a mini-batch.
> Generally, the incremental cost is negligible for a deep model.
> These methods have similar computational overhead, except for FactorVAE due to an extra discriminator network.
>
> Q4: Compared with FactorVAE and Cascade-VAE.
>
> A4: We compare CascadeVAE (beta\_max=10) and FactorVAR (gamma=20) in the following table and updated in Appendix A.8.
> Each tail repeats 5 times to get the mean and std scores.
> Similar to $\beta$-TCVAE, FactorVAE can not consistently outperform one hyperparameter on two datasets.
> Though CascadeVAE has good reconstruction fidelity, it cannot disentangle all factors properly.
> In this work, we aim to validate the workability of the proposed method, so we compare one representative method for each branch.
>
>
>
> |  dataset |    model   |      Recon.     |      MIG      |        ELBO       |
> |:--------:|:----------:|:---------------:|:-------------:|:-----------------:|
> | dSprites |  FactorVAE |  21.55±0.84 | 0.34±0.04 |  -46.05±2.24  |
> |          | CascadeVAE |  12.04±1.23 | 0.20±0.07 | -32.14 ± 1.29 |
> |          |   Dynamic  |  19.81±1.19 | 0.35±0.01 |  -37.83±1.17  |
> |          | beta-TCVAE |  73.04±3.41 | 0.29±0.10 |  -82.29±3.71  |
> |          |  beta-VAE  |  48.75±2.84 | 0.17±0.05 |  -61.17±3.13  |
> |          |    DeVAE   | 36.02±20.02 | 0.32±0.11 |   -51±22.26   |
> | Shapes3D |  FactorVAE |  18.48±2.28 | 0.38±0.28 |  -38.08±1.87  |
> |          | CascadeVAE |  14.84±1.98 | 0.46±0.11 |  -32.54±2.10  |
> |          |   Dynamic  |  29.70±4.15 | 0.54±0.04 |  -47.68±4.28  |
> |          | beta-TCVAE |  44.53±5.69 | 0.49±0.11 |  -60.01±6.29  |
> |          |  beta-VAE  |  34.95±2.34 | 0.42±0.18 |  -49.09±2.72  |
> |          |    DeVAE   | 46.80±13.97 | 0.52±0.10 |  -74.73±31.66 |
>
> Reference:
>
> 1. Jinkun Cao, Ruiqian Nai, Qing Yang, Jialei Huang, and Yang Gao. An empirical study on disentanglement of negative-free contrastive learning. In Alice H. Oh, Alekh Agarwal, Danielle Belgrave, and Kyunghyun Cho (eds.), Advances in Neural Information Processing Systems, 2022. URL https://openreview.net/forum?id=fJguu0okUY1.

---

> > ### Comment · Reviewer_8f7N · 2022-12-11
> > **Response to the authors**
> >
> > I thank the authors for answering all my questions and running the additional experiments. I think the paper has improved much in the revision. However, the paper lacks in two major ways:
> >
> > - The clarity of the paper is low in my opinion — the method and experiments sections need significant improvements for the paper to be easy to read and understand.
> > - The proposed method performs strictly worse than DynamicVAE and FactorVAE for dSprites and worse than DynamicVAE for Shapes3D. In various other cases, it’s unclear what the correct tradeoff between disentanglement and reconstruction is, so it’s hard to say which model would be a better option.
> >
> > Overall, in my opinion, a new disentanglement method should perform at least as well as the previous methods and/or clearly explain the applications/conditions where their method is more desirable over the previous methods, which this paper fails to do. Therefore, I am deciding to keep my rating.

---

> ### Author Response · Authors · 2022-11-15
> **Lamguage Improvement**
>
> Language improvements:
>
> - "DeVAE surpasses 2\% for $\beta$-TCVAE and 9\% for β-VAE."
>
> There is a mistake. It should be "DeVAE has a lower average reconstruction error on two datasets by 2\% for \btcvae{} and by 30\% for \betavae{}.", which is updated in Section 3.2.
> To show the reconstruction fidelity, we also demonstrate the samples of generated images of the best model with the highest MIG score for each method in Figure 8-10 in Appendix A.5.
>
> - "spreading the conflict of disentanglement and reconstruction over time and space"
>
>  Two targets, reconstruction and disentanglement, can not be optimized in one latent space and at a time for those based on IB.
> Previous IB-based methods apply a multi-stage or annealing strategy to optimize two goals in a different timeline.
> DeVAE creates several latent spaces, unlike these methods, and each latent space focuses on optimizing disentanglement or reconstruction.
> DeVAE is the first work to optimize them at the same time but in different spaces.
> We remove the confusing expression and update the description in the Abstract, Introduction, and Method parts.
>
> - “However, in this work, we get rid of calculating TC ..."
>
> We improve this sentence in the introduction:
> "However, the current TC estimations are not scaled to high dimensional problems, causing the low performance of BC-based
> methods in practice. In this work, instead of calculating TC, we leverage the information bottleneck
> (IB) (Tishby et al., 1999; Burgess et al., 2018) to promote disentanglement"
>
> - How are the hyperparameters for other methods chosen? Are the beta values chosen for DeVAE selected based on test performance or on a validation set?
>
> We did not use any validation set. We use an empirical choice of betas according to the information freezing points (IFP) [1].
> For example, beta=40 can filter factors orientation and shape, beta=10 can only filter factor orientation.
> Therefore, we add two extra layers with beta=10 and beta=40 to encourage the first layer for orientation and the second layer for shape.
> We supplement the explanation in Section 3.1.
>
> - What are the layer embeddings? They seem to be an important component of the method but are never explained.
>
> A layer embedding ${v}_i\in\mathbb{R}^{1\times D}$ is a learnable vector for the $i$th layer to identify the layer index for the decoder, like PosEmbed in ViT. The decoder takes a layer embedding and latent variables from one layer as inputs $\log{p_\theta ({x}|{z}_i,{v}_i)}$, see in Equation 2.
>
> Reference:
>
> 1. Jiantao Wu, Lin Wang, Bo Yang, Fanqi Li, Chunxiuzi Liu, and Jin Zhou. DEFT: distilling entangled factors by preventing information diffusion. Mach. Learn., 111(6):2275–2295, 2022. doi: 10.1007/s10994-022-06134-7. URL https://doi.org/10.1007/s10994-022-06134-7.

---

### Author Response · Authors · 2022-11-15
**General responses to all reviewers**

We are grateful for the valuable comments and suggestions from all reviewers.
In this revision, we have improved the expression and supplemented some experiments to support our claim.

Motivation and method: Previous incremental methods with only one latent space cannot optimize disentanglement and reconstruction simultaneously.
Our work proposes a novel decremental variational
autoencoder with hierarchy latent spaces to optimize multiple objectives in these layers, and all layers are connected by disentanglement-invariant transformations to optimize disentanglement jointly, termed DeVAE.
Specifically, the first layer aims to optimize the ELBO for reconstruction fidelity, and the subsequent layers having narrow information bottlenecks promote disentanglement.
As a result, DeVAE can optimize reconstruction and keep the constraint of disentanglement while training.

All modifications are colored blue in this revision. Here is the changelog:
1. In the **Abstract**, we modify the confusing expression "spreading the conflict over time" and the inaccurate description "best".
2. We improve the language of our motivation and method in **Section 1**.
3. We add details of DeVAE in **Figure 1 **, rewrite the method part in **Section 2.2**, and fix a problem in **Algorithm 1**.
4. We improve the math expression of $\tau$ and $p(z_i)$ in **Section 2.3 2.4**.
5. We improve the descriptions of experimental results in **Section 3.2**.
6. We add more visualizations to access DeVAE:
    - **Appendix A.5** shows the random sampling images.
    - **Appendix A.6** shows the change of mutual information along the layer for DeVAE.
7. We investigate the difference between DeVAE and DynamicVAE on high-dimensional latent spaces in  **Appendix A.7**.
8. We add the comparison with FactorVAE and CascadeVAE on dSprites in **Appendix A.8**.

---

### Decision · Program_Chairs · 2023-01-20

**Decision:**

Reject

**Justification For Why Not Higher Score:**

The issues of this papers are clear, and reviewer ratings are unanimously low.

**Justification For Why Not Lower Score:**

N/A

**Metareview: Summary, Strengths And Weaknesses:**

The paper presents a hierarchy of latent representations to gradually promote disentangled representations, where the loss for each latent layer has a KL-term and the weight monotonically increases with depth. The authors empirically demonstrate their idea on standard synthetic datasets, to show the proposed method is able to learn disentangled representations while preserving reconstruction fidelity, outperforming several previous methods.

Strength

The proposed method is intuitive, resembling prior work on increasing the KL weight (\beta) during training, but the authors propose to have multiple layers with increasing weight so the paper has some novelty. The experiments are somewhat comprehensive.

Weakness

The reviewers had severe concerns about the presentation of the paper.  On the other hand, while the intuitions are reasonable, there is no strong theoretical justification. Given there are multiple layers, the choice of \beta's may become tricky and the paper does not yet provide clear guidance on how to set them.